# THERE AND BACK AGAIN: ON THE RELATION BETWEEN NOISES, IMAGES, AND THEIR INVERSIONS IN DIFFUSION MODELS

## ABSTRACT

Denoising Diffusion Probabilistic Models (DDPMs) achieve state-of-the-art performance in synthesizing new images from random noise, but they lack meaningful latent space that encodes data into features. Recent DDPM-based editing techniques try to mitigate this issue by inverting images back to their approximated staring noise. In this work, we study the relation between the initial Gaussian noise, the samples generated from it, and their corresponding latent encodings obtained through the inversion procedure. First, we interpret their spatial distance relations to show the inaccuracy of the DDIM inversion technique by localizing latent representations manifold between the initial noise and generated samples. Then, we demonstrate the peculiar relation between initial Gaussian noise and its corresponding generations during diffusion training, showing that the high-level features of generated images stabilize rapidly, keeping the spatial distance relationship between noises and generations consistent throughout the training.

## 1 INTRODUCTION

Diffusion-based probabilistic models (DDPMs) (Sohl-Dickstein et al., 2015), have surpassed state-of-the-art solutions in many generative domains including image (Dhariwal & Nichol, 2021), speech (Popov et al., 2021)), video (Ho et al., 2022), and music (Liu et al., 2021) synthesis. Nevertheless, one of the significant drawbacks that distinguishes diffusion-based approaches from other generative models like Variational Autoencoders (Kingma & Welling, 2014), Flows (Kingma & Dhariwal, 2018), or Generative Adversarial Networks (Goodfellow et al., 2014) is the lack of implicit latent space that encodes training data into low-dimensional, interpretable representations. Several works try to mitigate this issue, treating as latent features internal data representations extracted from the denoising UNet model (Kwon et al., 2022), combining diffusion models with additional external models (Preechakul et al., 2021), or by seeking structure in the starting noise used for generations (Song et al., 2020).

The last example, introduced by (Song et al., 2020) with Denoising Diffusion Implicit Models (DDIM), led to the proliferation of methods grouped under the name of inversion techniques (Garibi et al., 2024; Mokady et al., 2023; Huberman-Spiegelglas et al., 2024; Meiri et al., 2023; Hong et al., 2024). The main idea behind those approaches is to use a diffusion model to predict the noise that can be added to the original or generated image. Applying this procedure several times allows tracing back the backward diffusion process and approximating the initial noise that results in the starting image. However, due to approximation error and biases induced by a trained denoising model, there are inconsistencies between initial Gaussian noise and the reversed so-called *latent* representation. While recent works try to improve the approximation and mitigate discrepancies between noise and latent, in this work, we propose to closely study the DDIM inversion procedure and highlight the relation between Gaussian noise, generated samples, and their latents.

First, we emphasize the main differences between initial Gaussian noise and latent codes. We locate the latent space between the initial Gaussian noise and generated samples, showing that DDIM inversion does not properly turn the image into noise. We show how this relation changes with time, highlighting the importance of early training steps in the formation of sample-to-latent mappings. Second, we move to the analysis of the relation between noise and samples. We show that it is

possible to correctly assign initial Gaussian noise to the generated sample with Euclidean distance. We further deepen this analysis to a non-stationary setup, showing that the mapping between noises and generations emerges at the very beginning of the diffusion model training.

The main contribution of this work can be summarized as follows:

- We show that reverse DDIM produces latent representations that are not standard multivariate Gaussian, creating a gap between the diffusion models' theory and practice.

- We study how this relation changes with training and show that improving the generative capabilities of the model does not improve the accuracy of reverse DDIM.

- We show that the relation between images and noises is defined by the simple L2 distance at the early stage of the training.

## 2 BACKGROUND

The training of diffusion models consists of forward and backward diffusion processes, where in the context of Denoising Diffusion Probabilistic Models (DDPMs), the former one with training image $x_0$ and $\{\beta_t\}_{t=1}^T$ being some variance schedule, can be expressed as

$$x_t = \sqrt{\bar{\alpha}_t} x_0 + \sqrt{1 - \bar{\alpha}_t} \epsilon_t, \tag{1}$$

with $\alpha_t = 1 - \beta_t$, $\bar{\alpha}_t = \prod_{s=1}^t \alpha_s$, and $\epsilon_t \sim N(0, I)$.

In the backward process, the noise is gradually removed starting from a random noise $x_T \sim N(0, I)$,

$$x_{t-1} = \sqrt{\frac{\bar{\alpha}_{t-1}}{\bar{\alpha}_t}}(x_t - \sqrt{1 - \bar{\alpha}_t} \cdot \epsilon_\theta(x_t, t, c)) + \sqrt{1 - \bar{\alpha}_{t-1} - \sigma_t^2} \cdot \epsilon_\theta(x_t, t, c) + \sigma_t z_t, \tag{2}$$

with $\sigma_t = \eta \beta_t (1 - \bar{\alpha}_{t-1})/(1 - \bar{\alpha}_t)$ being a variance schedule, $\epsilon_\theta(x_t, t, c)$ being the output of a network trained to remove noise, and $z_t \sim N(0, I)$.

While for $\eta = 1$, a non-deterministic DDPM model is used, setting $\eta = 0$ removes the random component from the equation 2, making it a Denoised Diffusion Implicit Model (DDIM), characterized by a deterministic mapping from noise space $x_T$ to image space $x_0$.

By removing the stochasticity of the DDPM sampling, we can additionally reverse deterministically the direction of the backward diffusion process and encode images back to the original noise space. The DDIM inversion is obtained by rewriting equation 2 as:

$$x_t = \sqrt{\frac{\bar{\alpha}_t}{\bar{\alpha}_{t-1}}} x_{t-1} + \left(\sqrt{1 - \bar{\alpha}_t} - \frac{\sqrt{\bar{\alpha}_t - \bar{\alpha}_t \bar{\alpha}_{t-1}}}{\sqrt{\bar{\alpha}_{t-1}}}\right) \cdot \epsilon_\theta(x_t, t, c) \tag{3}$$

However, due to circular dependency on $\epsilon_\theta(x_t, t, c)$, DDIM inversion approximates this equation by assuming linear trajectory with direction to $x_T$ in $t$-th step being same as in $(t-1)$-th step, i.e.,

$$\epsilon_\theta(x_t, t, c) \approx \epsilon_\theta(\mathbf{x_{t-1}}, t, c). \tag{4}$$

While such approximation is often sufficient to obtain good reconstructions of images, it introduces the error that depends on the difference $(x_t - x_{t-1})$, which can be detrimental for models that leverage a few diffusion steps or use the classifier-free guidance Ho & Salimans (2021); Mokady et al. (2023). In this work, we empirically study the consequences of this approximation error.

We will be interested in studying the relations between the following three objects:

- Gaussian noise variable, $\mathbf{x^T}$, used to generate an image through a diffusion process.

- Image sample, $\mathbf{x^0}$, the outcome of the generation process produced by the Diffusion Model, i.e., the result of going through denoising stages, starting from $t = T$ and ending on $t = 0$.

- Latent variable, $\hat{\mathbf{x}}^\mathbf{T}$, the result of starting from $\mathbf{x^0}$ and applying $T$ steps of a reversed DDIM generation process, see equation 4.

## 3 RELATED WORK

**Diffusion models inversion techniques**   Thanks to the reversible sampling procedure, Denoising Diffusion Implicit Models are often employed in tasks such as inpainting (Zhang et al., 2023), image (Su et al., 2022; Kim et al., 2022; Hertz et al., 2022), video (Ceylan et al., 2023) or speech (Deja et al., 2023a) edition. However, the baseline DDIM approach is based on the assumption that the prediction of the noise removed from the image in the $t$-th backward diffusion step closely approximates the noise of the $(t − 1)$-th step. This assumption is not always true, and recent works try to mitigate the discrepancies of such approximation. In particular, Renoise (Garibi et al., 2024) iteratively improves the prediction of added noise using the predictor-corrector technique. This allows to closely estimate the original Gaussian noise for a given image, even with fewer diffusion steps. Several works aim to reverse the diffusion process in text-to-image models. In such a case, the prompt selection significantly influences the final latent. To mitigate this issue, Null-text inversion method (Mokady et al., 2023) extends the DDIM inversion with optimized pivotal noise vectors and additional Null-text optimization technique, where unconditional textual embeddings employed by classifier free-guidance (Ho & Salimans, 2021) are optimized in order to reduce the reconstruction error. Other work (Huberman-Spiegelglas et al., 2024) proposes an alternative DDPM noise space, where noise maps do not have a normal distribution, yet it enables better image editing capabilities and perfect reconstructions. On the other hand, a regularized Newton-Raphson inversion method (Meiri et al., 2023) formulates the inversion process as solving an implicit equation using numerical techniques, achieving faster and higher-quality reconstructions with prompt-aware adjustments, enabling improvements in image interpolation, and boosting models' diversity. Furthermore, exact inversion methods for DPM-solvers (Hong et al., 2024) mitigate the errors introduced by classifier-free guidance by leveraging gradients, boosting both the image and noise reconstruction.

**Latent space in diffusion models**   One of the significant drawbacks of diffusion-based generative models compared to approaches such as VAEs, Flows, or GANs is the lack of meaningful latent space that encodes features of the generated samples. Several approaches try to mitigate this issue. Kwon et al. (2022) show that so-called *h-features*, which are the activations located inside of the U-Net model used as a diffusion decoder, can be used as meaningful representations providing space for semantically coherent image manipulation. This idea is further extended by Park et al. (2023), where the authors show that we can calculate the pullback metric that directly associates h-features with the original image space. Several works show that we can directly benefit from such features in downstream tasks such as image segmentation (Baranchuk et al., 2021; Tumanyan et al., 2023; Rosnati et al., 2023), image correspondence (Luo et al., 2024) or classification (Deja et al., 2023b).

**Noise-to-image mapping in diffusion models**   Contrary to latent variable models such as VAEs, the forward diffusion process that maps images to the Gaussian noise is a parameter-free process that can be understood as hierarchical VAE with pre-defined unlearnable encoder Kingma et al. (2021). However, several works show interesting properties resulting from the training objective of DDPMs and score-based models. Kadkhodaie et al. (2024) show that due to inductive biases of denoising models, different DDPMs trained on similar datasets converge to almost identical solutions. This idea is further explored by Zhang et al. (2024), where the authors show that even models with different architectures converge to the same score function and, hence, the same noise-to-generations mapping. This mapping itself is further analyzed by Khrulkov & Oseledets (2022), where the authors show that the encoder map coincides with the optimal transport map for common distributions. In this work, we extend this analysis further by empirically validating that examples are generated from the closest random noise, even for more complex distributions.

## 4 EXPERIMENTS

### 4.1 EXPERIMENTAL DETAILS

For the training, we follow Nichol & Dhariwal (2021) and train two unconditional pixel-space diffusion models on ImageNet and CIFAR10 datasets. In our studies we use three diffusion models, a DDPM-based Dhariwal & Nichol (2021) model trained on CIFAR10 and ImageNet, and a Latent Diffusion Model (LDM) trained on CelebA dataset. The CIFAR10 model was trained for 500K steps, while the ImageNet one for 1.5M steps. Both DDPM models use 4K diffusion steps during

their training. For both diffusion models trained on the CIFAR10 and ImageNet datasets, we investigate the noising and denoising process with a number of diffusion steps $T$ varying from $50$ up to $4K$. For our experiments, we average our metrics over 1K samples generated from 3 models trained with 3 random training seeds.

## 4.2 Noise $\neq$ Latent

Numerous methods employ reversed DDIM in applications such as image editing or inpainting. In this case, the underlying assumption is that by *encoding* the image back with a denoising decoder, we can obtain the original *noise* that can be used to reconstruct the original image. However, as noticed by recent works Garibi et al. (2024); Parmar et al. (2023); Hong et al. (2024), the latents created with DDIM inversion ($\hat{\mathbf{x}}^{\mathbf{T}}$) often do not follow standard multivariate Gaussian distribution, creating a gap between the promise of diffusion models' theory and practice. We study the implications of this fact and show its far-reaching consequences.

We visualize this phenomenon in Figure 1 for three models, showing the latent or its difference from the generated image. For simpler ones trained with smaller datasets, such as CIFAR10, we can observe clear structures of original images in the inverted latents, as presented in Figure 1 (C). However, even for more complex datasets, we can highlight the inversion error by plotting the image difference between the latent and the noise, as presented for DDPM trained on ImageNet and LDM trained on CelebA (Figure 1 (A, B)). We can observe that the highest inversion error can be observed in the areas of large monotonic surfaces, where more high-frequency noise needs to be added.

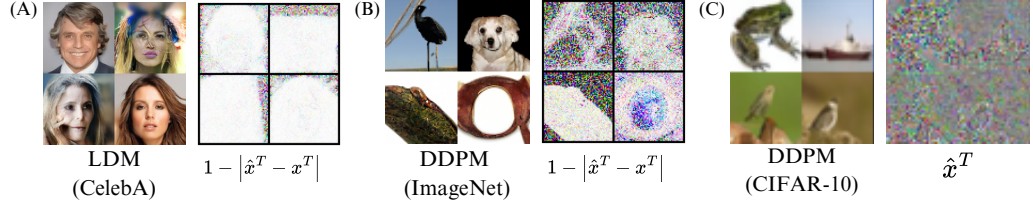

Figure 1: Visualization of samples and latent images from three diffusion models. For smaller DDPM model trained on CIFAR10 (C) we can observe original image structure in the latent calculated with reverse DDIM. We can observe similar structures on image differences from larger models (A and B).

To measure the extent of this effect, in Table 1, we show the values of the top-10 Pearson correlation coefficients measured between individual pixels of either initial noise ($\mathbf{x}^{\mathbf{T}}$), latent ($\hat{\mathbf{x}}^{\mathbf{T}}$), or samples ($\mathbf{x}^{\mathbf{0}}$). We can observe correlated pixels in the latents, which further supports the claim that they do not come from the standard multivariate Gaussian distribution. This is especially true for models trained on smaller datasets such as CIFAR-10 and almost invisible for latent models.

|  | DDPM (CIFAR10) | DDPM (ImageNet) | LDM (CelebA) |
|---|---|---|---|
| Noise ($\mathbf{x}^{\mathbf{T}}$) | $0.159 \pm 0.003$ | $0.177 \pm 0.007$ | |
| Latent ($\hat{\mathbf{x}}^{\mathbf{T}}$) | $0.462 \pm 0.009$ | $0.219 \pm 0.006$ | $0.179 \pm 0.008$ |
| Sample ($\mathbf{x}^{\mathbf{0}}$) | $0.986 \pm 0.001$ | $0.966 \pm 0.001$ | $0.904 \pm 0.005$ |

Table 1: Top-10 correlation coefficients in random Gaussian noise vs. latent. We can observe that latents created with reverse DDIM have correlated pixel values. This is especially visible for smaller DDPM models and almost not noticeable for LDM.

## 4.3 What is the location of the latent variable?

Given the results of the previous analysis, we pose the next question: *what is the location of the latent variables calculated with the reverse DDIM procedure?* We show that the DDIM-generated latent ($\hat{\mathbf{x}}^{\mathbf{T}}$) is located along the trajectory $x_t$ between the Gaussian noise ($\mathbf{x}^{\mathbf{T}}$) and the generated sample ($\mathbf{x}^{\mathbf{0}}$), and discuss how this relation changes with different characteristics of diffusion models.

In the first experiment, shown in Figure 2, we calculate the angles between noises, samples, and latents. We average the angles across 1K samples from three different models and plot the triangles in 2D space. We can observe that in each case, the angle located at the image's vertex, $\angle \mathbf{x^0}$, is acute but never zero. On the other hand, the angle located at the vertex representing latents, $\angle \hat{\mathbf{x}}^\mathbf{T}$, is always obtuse. This leads to the conclusion that due to the imperfect approximation of the reverse DDIM procedure, latents are located along the trajectory $x_t$ of the generated image.

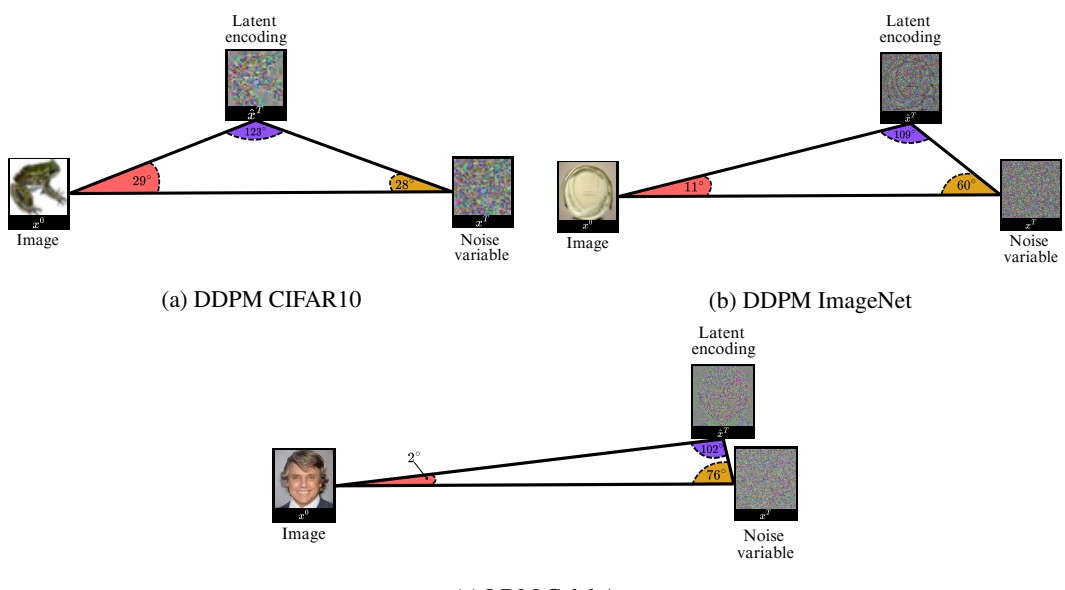

(a) DDPM CIFAR10           (b) DDPM ImageNet

(c) LDM CelebA

Figure 2: Visualisation of the most probable relation between random Gaussian noises, their corresponding samples, and latents recovered with reverse DDIM procedure for three different models.

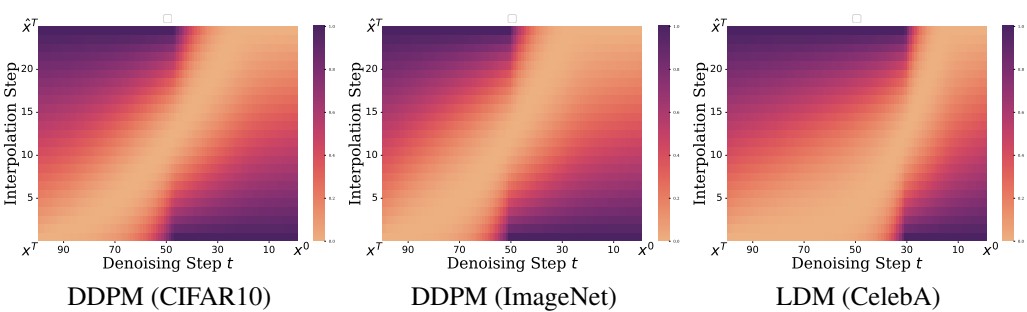

DDPM (CIFAR10)       DDPM (ImageNet)       LDM (CelebA)

Figure 3: Distances between intermediate steps of backward diffusion process, and the interpolated points between initial noise $\mathbf{x^T}$ and the inverted latents $\hat{\mathbf{x}}^\mathbf{T}$. We can observe that independently of the model, the consecutive intermediate generations along the sampling trajectory initially get closer to the latent variable up to the point where after approximately 50-70% the generations pass the latent. This aligns with the visualization of the most rpboable relations in Figure 2.

To approximate the exact location of the latents more closely, we analyze the distance between different steps of the backward diffusion process and the Noise-Latent side of the triangle, i.e., the interval $\mathbf{x^T}\hat{\mathbf{x}}^\mathbf{T}$, see Figure 3. Each pixel, with coordinates $(t, \lambda)$, is colored according to the L2 distance between the intermediate step of trajectory $x_t$ and the corresponding interpolation step, i.e. $\|(1 - \lambda)\mathbf{x^T} + \lambda\hat{\mathbf{x}}^\mathbf{T} - x_t\|_2$. Figure 3 shows that while moving from the random Gaussian noise ($\mathbf{x^T}$) towards the final sample ($\mathbf{x^0}$), the intermediate steps $x_t$ are getting closer to the latent ($\hat{\mathbf{x}}^\mathbf{T}$), which is visually represented by the light-colored area in the upper right corner of the plot. This also demonstrates that from some time onwards, say $t_0$, the trajectory $x_t$ closest point to the interval $\mathbf{x^T}\hat{\mathbf{x}}^\mathbf{T}$ is the right endpoint $\hat{\mathbf{x}}^\mathbf{T}$. This has non-trivial consequences in the light of $\hat{\mathbf{x}}^\mathbf{T}$ featuring a lot of structure (see Section 4.2), in the form of an unprecedented amount of structure between $x_t, \mathbf{x^0}, \hat{\mathbf{x}}^\mathbf{T}$

unaccounted by theory. Failing to realize this implication could lead to incorrect reasoning and spurious discoveries. The same trend can be observed across all three evaluated models.

Finally, we show that the situation is persistent across the training process; see Figure 4. Hypothetically, the value of all three triangle angles shown in Figure 2 could fluctuate during the training process. However, both the angle adjacent to the image $\angle \mathbf{x^0}$ and the distance between the image and noise quickly converge to a certain value that remains constant through the rest of the training. This observation brings two main conclusions: (1) The relation between noises, latents, and samples is defined at the early stage of the training, and (2) The inverse DDIM method does not benefit from the prolonged training time of the diffusion model.

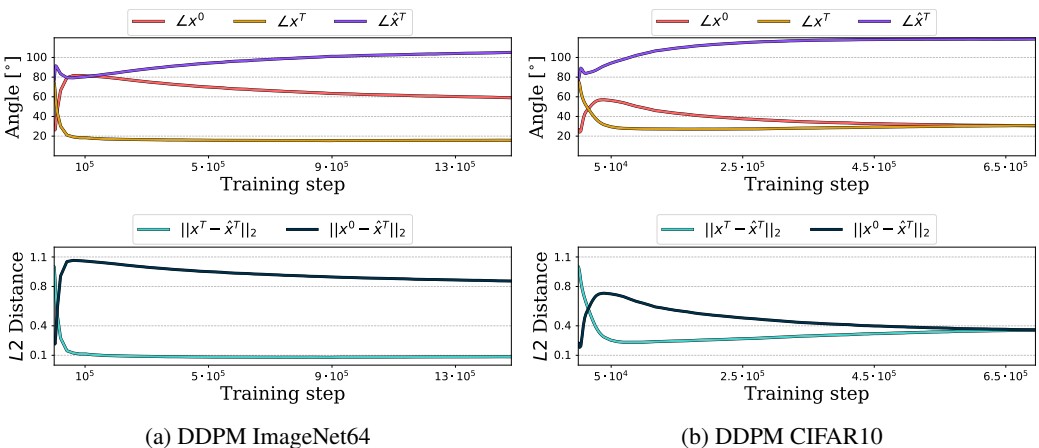

(a) DDPM ImageNet64          (b) DDPM CIFAR10

Figure 4: We generate 1000 samples from the final model and revert them to the corresponding latents using an intermediate training checkpoint saved after a given number of training steps.

### 4.4 NOISE-TO-SAMPLE MAPPING

As indicated by previous works (Kadkhodaie et al., 2023; Zhang et al., 2024), diffusion models converge to the same mapping between the random Gaussian noise ($\mathbf{x^T}$) and the generated images ($\mathbf{x^0}$) independently on the random seed, parts of the training dataset, or even the model architecture. In this work, we investigate this phenomenon further and study the nature of mapping between noises and samples and how it changes during training.

| T | CIFAR10 (DDPM) | | ImageNet (DDPM) | | CelebA (LDM) | |
|---|---|---|---|---|---|---|
| | $x^0 \rightarrow x^T$ | $x^T \rightarrow x^0$ | $x^0 \rightarrow x^T$ | $x^T \rightarrow x^0$ | $x^0 \rightarrow x^T$ | $x^T \rightarrow x^0$ |
| 10 | $90.3_{\pm 6.3}$ | $94.0_{\pm 2.6}$ | $99.4_{\pm 0.0}$ | $100_{\pm 0.0}$ | $100_{\pm 0.0}$ | $100_{\pm 0.0}$ |
| 100 | $98.9_{\pm 1.2}$ | $50.4_{\pm 1.9}$ | $100_{\pm 0.0}$ | $59.0_{\pm 7.1}$ | $100_{\pm 0.0}$ | $100_{\pm 0.0}$ |
| 1000 | $99.1_{\pm 1.0}$ | $46.8_{\pm 3.0}$ | $99.8_{\pm 0.2}$ | $44.6_{\pm 6.3}$ | $100_{\pm 0.0}$ | $100_{\pm 0.0}$ |
| 4000 | $99.1_{\pm 1.0}$ | $46.4_{\pm 3.0}$ | $99.5_{\pm 0.3}$ | $43.3_{\pm 6.7}$ | - | - |

Table 2: Accuracy on assigning noise to the corresponding generated image ($\mathbf{x^0} \rightarrow \mathbf{x^T}$) and vice-versa ($\mathbf{x^T} \rightarrow \mathbf{x^0}$). In DDPMs, we are able to correctly select the original noise for a given sample by calculating the L2 distances, while the reverse assignment is only valid for a small number of diffusion steps. In LDM, we can correctly predict assignments in both directions.

To that end, we first generate 1K samples from random Gaussian noises and try to predict which noise ($\mathbf{x^T}$) was used to generate which sample ($\mathbf{x^0}$) and vice-versa. As presented in Table 2, we show that we can accurately assign the images to the corresponding noises ($\mathbf{x^0} \rightarrow \mathbf{x^T}$) according to the smallest L2 distance criterion. This is especially true for the higher number of diffusion timesteps, where for all models, we achieve over 99% accuracy. The situation changes when trying to assign the noise to the corresponding generated image ($\mathbf{x^T} \rightarrow \mathbf{x^0}$). We can observe high accuracy with a low number of generation timesteps ($T = 10$), but the results deteriorate quickly with the

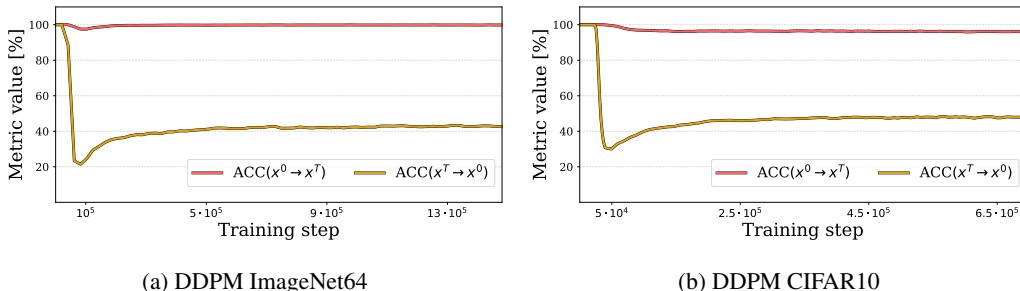

(a) DDPM ImageNet64        (b) DDPM CIFAR10

Figure 5: Accuracy of assigning initial noise given the generated sample $(\mathbf{x^0} \to \mathbf{x^T})$ and sample given the initial noise $(\mathbf{x^T} \to \mathbf{x^0})$ when training the diffusion model. We can observe that from the very beginning of training, we can assign initial noise with a simple L2 distance, while the accuracy of the reverse assignment rapidly drops.

increase of this parameter. The reason for this is that greater values of $T$ allow the generation of a broader range of images, including the ones with large plain areas of low pixel variance. Such generations turn out to be close to the majority of initial Gaussian noises. We provide examples of such generations in the Appendix A.1. Notably, we do not observe such behavior in the LDM model, where final generations are well normalized through the KL-divergence applied to the latent space of the LDM's autoencoder.

To further analyze the nature of this property, we show how those metrics change when sampling generations from intermediate checkpoints of the diffusion models' training; see Figure 5. We can observe that the distance between noises and latents accurately defines the assignment of initial noises given the generated samples $(\mathbf{x^0} \to \mathbf{x^T})$ from the beginning of the training till the end. At the same time, the accurate reverse assignment $(\mathbf{x^T} \to \mathbf{x^0})$ can only be observed at the beginning of the training, when generations have not yet been properly formed.

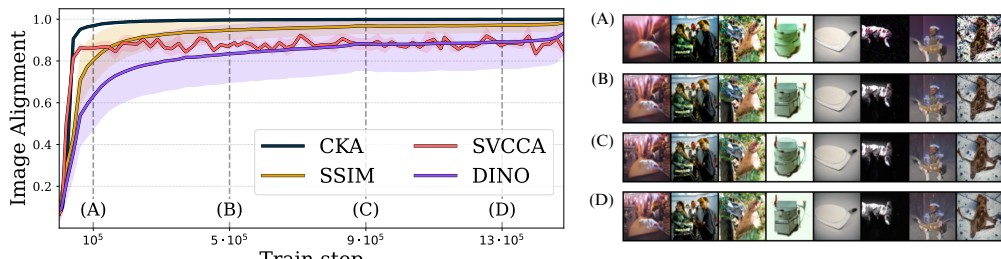

Figure 6: Similarity of the generations sampled from the same random noise at different stages of diffusion model's training to the final outputs for DDPM and ImageNet. Already after a few epochs, model learns the mapping between Gaussian noise and generations. Prolonged training improves the quality of samples, adding high-frequency features, without changing their content. This can be observed through different image alignment metrics (left) and visual inspection (right).

To further measure how the relation between noises and samples change when training the model, for each training step $n \in \{1 \dots 500K\}$ for CIFAR10 and $\{1 \dots 1.5M\}$ for ImageNet we generate 1K samples $\{\mathbf{x}^{\mathbf{0}}_{i,n}\}_{i=1}^{1000}$ from *the same random noise* $\mathbf{x}^{\mathbf{T}}_{\text{fixed}} \sim N(0, I)$, and compare them with generations obtained for the fully trained model. We present the visualization of this comparison in Figure 6 using CKA, DINO, SSIM, and SVCCA as metrics. We notice that image features rapidly converge to the level that persists until the end of the training. This means that prolonged learning does not significantly alter how the data is assigned to the Gaussian noise after the early stage of the training. It is especially visible when considering the SVCCA metric, which measures the average correlation of top-10 correlated data features between two sets of samples.

We can observe that this quantity is high and stable through training, showing that generating the most important image concepts from a given noise will not be affected by a longer learning pro-

cess. For visual comparison, we plot the generations sampled from the model trained with different numbers of training steps in Figure 6 (right).

## 5 CONCLUSIONS

In this work, we empirically study the relation between initial Gaussian noise, generated samples, and their latent representations calculated with the inverse DDIM technique. First, we show that the error in the approximation of the previous noise in DDIM leads to representations located next to the generation trajectory between starting samples and their true noises. Moreover, prolonged diffusion training does not affect this property, as the accuracy of DDIM inversion does not improve in time. Then, studying the relation between the generated samples and Gaussian noise, we show that we can accurately assign the initial noise of the given generation with a simple L2 distance. We also demonstrate that this behavior emerges at the very beginning of the diffusion models training. Our experiments lead to the conclusion that the initial part of the diffusion model's training is responsible for building the relation between the initial Gaussian noise, final generations, and their inverted representations.

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

## A  APPENDIX

### A.1  MISTAKES IN NOISE TO SAMPLE MAPPING

In Figures 7 and 8 we show examples of images that are not properly assigned to their initial noises. We can observe that those images are characterised by low variance of pixels.

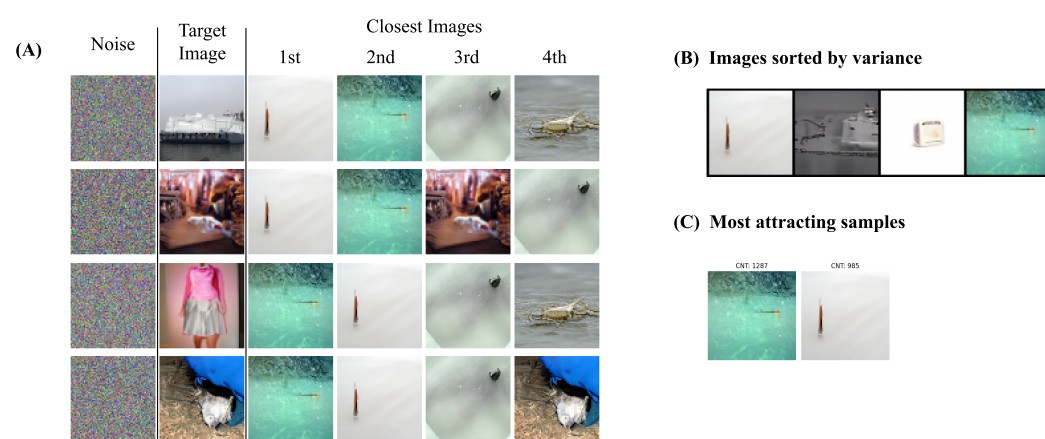

Figure 7: Examples assigned to the wrong initial noises for CIFAR10 datasets

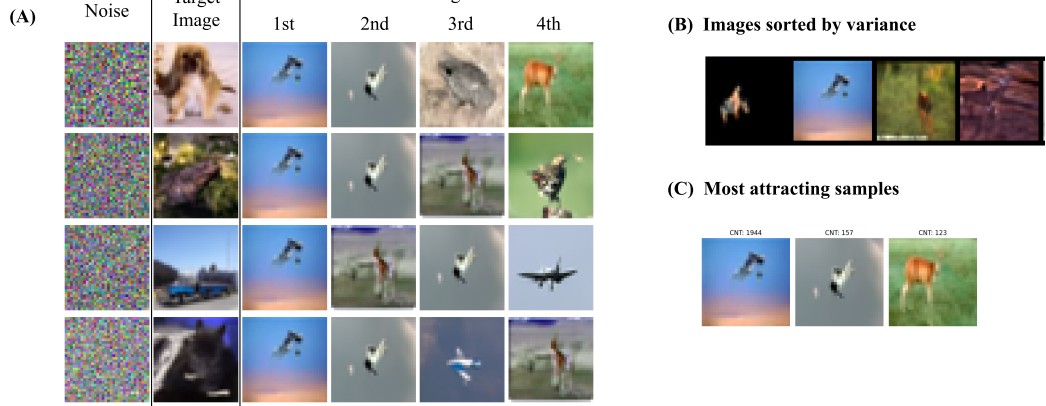

Figure 8: Examples assigned to the wrong initial noises for CIFAR10 datasets