# OpenReview forum: "There and Back Again: On the relation between noises, images, and their inversions in diffusion models"
_ICLR.cc/2025/Conference — Submitted to ICLR 2025_

### Official Review · Reviewer_khzk · 2024-10-31

**Soundness:** 2
**Presentation:** 2
**Contribution:** 4
**Rating:** 8
**Confidence:** 4

**Summary:**

This paper analyzes intriguing properties of the relationship between the triplet (image, noise, latent) in diffusion models, where “latent” refers to the noise computed by the inverted diffusion process in deterministic diffusion models (DDIM). Even if from a theoretical point of view, noise and latent should match, in practice this does not happen due to approximations required to practically implement the inverted diffusion process. This analysis is performed in two stages: first of all, the authors shows that the latent variable is located next to the trajectory mapping noise to the generated image, by analyzing both the angles of the triangle with vertices (image, noise, latent), and by computing the distance of each element x_t of the trajectory with the edge connecting the noise to the latent. Moreover, they show that this behavior emerges at the beginning of the training, and never changes as the training advances. In the second stage, the authors argue that the mapping between noise and the generated image is “predictable”, in the sense that the noise gets mapped to the closest feasible generated data, measured in L2 loss. Again, they show that this behavior emerges at the very beginning of the training.

**Strengths:**

The idea presented in the paper is original, as not many authors analyzed the relationship that occurs between the noise that generates an image and the latent encoding of the image itself in deterministic diffusion models. Due to the nowadays popularity of DDIM, I also believe that this work is significant for the scientific community, as it describes intruguing properties of these models.

**Weaknesses:**

I believe the paper has a few aspects that require to be improved to allow for a precise decision. In particular:
- the description of the experiments is too hasty and does not go enough into the details, making it very hard to understand. In particular, the whole paper is very confusing in the distinction between DDPM and DDIM. In Section 2, the authors define DDPM as the model obtained by setting $\eta = 1$ in the definition of $\sigma_t$ in Equation (2), while DDIM is the model obtained by setting $\eta = 0$, and its reverse process is fully deterministic, once $x_T$ is given. Clearly, the inversion map, mapping $x_0$ to its latent, is only defined in the DDIM setting. In Section 4, however, the authors continuously interchange the names DDPM and DDIM, making it very hard to understand which model they are using. For example, both Table 1 and Figure 1 uses the name “DDPM” to indicate the models, while the first paragraph of Section 4.2 refers to them as “DDIM”. Therefore, I suggest the authors to rewrite this section by paying more attention to the definitions of DDIM and DDPM. Therefore, I suggest the authors to:
   1. clearly define DDPM and DDIM early in the paper and consistently use these terms throughout.
   2. explicitly state which model (DDPM or DDIM) is being used in each experiment in Section 4.
   3. explain how the inversion process is applied to DDPM models if that is indeed what they are doing.
- in Section 4.4, they refer to “the smallest L2 criterion” without introducing it. Moreover, the obtained results are confusing to me since I didn't expect a discrepancy in the accuracy of $x_0 \to x_T$ vs $x_T \to x_0$. In a final version of this work, I expect the authors to:
   1. Formally define the "smallest L2 criterion" when it's first mentioned.
   2. Explain the classification process using this criterion in more detail.
   3. Address the apparent discrepancy between the accuracies of $x_0$ -> $x_T$ and $x_T$ -> $x_0$, given the symmetry of the L2 norm.
- In line 348, after discussing the accuracy of the metrics between image and noises, the authors say “we can observe that the distance between noise and latents accurately defines…”. Note that, in this section, the latents were not considered, since all the experiments were performed on images and noises. Therefore I do not understand what they want to say with this sentence. In general, I suggest the authors to re-check Section 4 to correct the errors and improve the readability, better clarifying all the steps they performed. Please note that the length of the paper is at most 10 pages EXCLUDING the citations, therefore you still have 3 pages left, which you can use to expand the description of the experimental section.

**Minor Comments.**

-	In line 092, you say “DDIM inversion approximates this equation by assuming linear trajectory”. Explain these in more detail, since it is not clear to me why this assumption implies Equation (4).
-	In Equation (4), remove the bold from $x_{t-1}$ to be coherent with the notation employed in the paper.
-	In line 107, the referred equation should be Equation (3).
-	In the related works, I believe you should also consider at least the ODE inversion paper by Song et al. (i.e. the “probability flow” paper), and also Asperti et al., 2023 (title: Image embedding for denoising generative models), which introduces the DDIM inversion through neural network training.
-	In the list of models with “meaningful latent space encoding”, the authors considers also GAN. I believe this is not the case, since GAN has a similar behavior as DDIM, i.e. there is no meaningful nor explicit latent space, and it has to be recovered by techniques that are very similar to the ones used to invert DDIM, like GAN inversion.
-	I believe lines 158-159 should be postponed since they refer to the “two models” that you introduce in lines 159-160.
-	In lines 215 and 221 you say that the latent is located along the trajectory, while you actually show that it is next to the trajectory.

**Questions:**

I included a few questions in the "Weakness" section

---

> ### Author Response · Authors · 2024-12-03
> **Response to the Review (1/2)**
>
> We thank the Reviewer for the valuable feedback and their recognition of its originality and importance.
>
> > The description of the experiments is too hasty and does not go enough into the details, making it very hard to understand. In particular, the whole paper is very confusing in the distinction between DDPM and DDIM. [...] Therefore, I suggest the authors to:
> > 1. clearly define DDPM and DDIM early in the paper and consistently use these terms throughout.
> > 2. explicitly state which model (DDPM or DDIM) is being used in each experiment in Section 4.
> > 3. explain how the inversion process is applied to DDPM models if that is indeed what they are doing.
>
> We appreciate the Reviewer's pointing out this confusion. We agree this is misleading; hence, we want to clarify it in this comment. We use the DDIM sampler in the diffusion model inference process for all the experiments in the paper. This assumption allows us to get deterministic generations from the given noise and approximate this noise from generations with the DDIM inversion method. However, the models were trained with the standard DDPM schedulers, so we used those two terms alternately, and following [1], we used this term to distinguish pixel-space models that we called DDPMs and latent models (LDMs). We agree that this term is inacurate, so we will change the names of these models to "Pixel DMs" in the revised version of our paper.
>
>
> > In Section 4.4, they refer to “the smallest L2 criterion” without introducing it. Moreover, the obtained results are confusing to me since I didn't expect a discrepancy in the accuracy of $x_0\rightarrow x_T$ vs $x_T \rightarrow x_0$. In a final version of this work, I expect the authors to:
> > 1. Formally define the "smallest L2 criterion" when it's first mentioned.
> > 2. Explain the classification process using this criterion in more detail.
>
> Thank you for pointing out that the initial version of the submission lacked a thorough explanation of this aspect. For calculating a distance between two objects in our experiments, we use the $L_2$ norm (euclidean distance) between two images/noises/latents calculated as follows: ${||x-y||}_2 = \sqrt{\Sigma_c\Sigma_i\Sigma_j (x_{c,i,j}-y_{c,i,j})^2}$.
>
> For the classification experiment in Table 2, we assume a setup where we have $N$ inputs, called, in the paper, initial Gaussian noises, and their $N$ corresponding diffusion model outputs, called image generations. When assigning images to the noises, we iterate over all the $N$ noises, and for each one, we calculate its distance to all the $N$ generations. We assign the images to noises by choosing the one with lowest $L_2$ distance. For the accuracy metric, we check if, for a given noise, the predicted image is the same as the one that results from denoising this particular noise with DDIM). For the reverse problem (assignment of noises to images), the setup is the same, but we iterate over the $N$ image generations and, for each image, calculate its $L_2$-distances to all the noises and choose the one to which a distance is lowest. Please note that that even though the L2 distance is symetrical, the assignment of the closest image/noise is not the same due to the one-directional many-to-one relation (e.g. There might be several noises pointing towards the same closest image).
>
> > 3. Address the apparent discrepancy between the accuracies of $x_0\rightarrow x_T$ and $x_T\rightarrow x_0$, given the symmetry of the L2 norm.
>
> While the $L_2$ norm is symmetrical, the problem here is the many-to-one relation that we perform. When assigning images to the initial noises, there are singular generations (with large plain areas) located close to the mean of the random Gaussian noise in the set of generated images. Such generations tend to be the closest (in $L_2$-Norm) for the majority of the noises in our experiments. We show examples of such wrong assignments in Figure 7 (A) in the Appendix. In (C), we present the singular generations that lead to incorrect noise-to-image classification, along with the number of noises for which they are the closest. In (B), we sort images used in the experiment by the variance of pixels and show four with the lowest one. We observe that the set of singular generations leading to misclassification overlaps with lowest-variance generations.
>
> > In line 348, after discussing the accuracy of the metrics between image and noises, the authors say “we can observe that the distance between noise and latents accurately defines…”. Note that, in this section, the latents were not considered, since all the experiments were performed on images and noises. Therefore I do not understand what they want to say with this sentence.
>
> Thank you to the reviewer for pointing out this error. We confirm that in the sentence we meant "the distance between the noises and their corresponding generations".

---

> > ### Author Response · Authors · 2024-12-03
> > **Response to the Review (2/2)**
> >
> > > In line 092, you say “DDIM inversion approximates this equation by assuming linear trajectory”. Explain these in more detail, since it is not clear to me why this assumption implies Equation (4).
> >
> > As denoted in Equation (3) in the initial submission, to perform the exact DDIM inversion and obtain a noisier latent $x_t$ from a less noisy latent $x_{t-1}$, we would need the diffusion model's output for the latent we aim to obtain, $\epsilon_{\theta}(x_t, t, c)$. However, determining this output is infeasible due to the circular dependency on $x_t$.
> >
> > To address this, the DDIM inversion assumes a **local** linear trajectory in the latent space. The output of the noise-prediction model, $\epsilon_{\theta}(x_t, t, c)$, can be interpreted as a vector representing the direction from $x_t$ to $x_{t-1}$ during the diffusion denoising process. By swapping $\epsilon_{\theta}(x_t, t, c)$ with $\epsilon_{\theta}(x_{t-1}, t, c)$ in Equation (3), we approximate (locally) that the direction from $x_t$ to $x_{t-1}$ is the same as the direction from $x_{t-1}$ to $x_{t-2}$. Mathematically, such approximation implies that $x_t - x_{t-1} \approx x_{t-1} - x_{t-2}$.
> >
> > > In Equation (4), remove the bold from to be coherent with the notation employed in the paper.
> > > In line 107, the referred equation should be Equation (3).
> > > I believe lines 158-159 should be postponed since they refer to the “two models” that you introduce in lines 159-160.
> > > In lines 215 and 221 you say that the latent is located along the trajectory, while you actually show that it is next to the trajectory.
> >
> > We thank the Reviewer for these editorial suggestions. We will apply them in the final version of our paper.
> >
> > > In the related works, I believe you should also consider at least the ODE inversion paper by Song et al. (i.e. the “probability flow” paper), and also Asperti et al., 2023 (title: Image embedding for denoising generative models), which introduces the DDIM inversion through neural network training.
> >
> > Thank you for suggesting additional related works, we agree that they should be included in the appropriate section, and we will describe them in the paper.
> >
> > > In the list of models with “meaningful latent space encoding”, the authors considers also GAN. I believe this is not the case, since GAN has a similar behavior as DDIM, i.e. there is no meaningful nor explicit latent space, and it has to be recovered by techniques that are very similar to the ones used to invert DDIM, like GAN inversion.
> >
> > Thank you for pointing out this misleading claim. We can see that this statement indeed introduced a lot of confusion, among several reviewers, therefore we decided to remove it.
> >
> > **References:**
> >
> > [1] Nichol, Alexander Quinn, and Prafulla Dhariwal. "Improved denoising diffusion probabilistic models." International conference on machine learning. PMLR, 2021.
> >
> > [2] Dhariwal, Prafulla, and Alexander Nichol. "Diffusion models beat gans on image synthesis." Advances in neural information processing systems 34 (2021): 8780-8794.

---

### Official Review · Reviewer_8i5c · 2024-11-04

**Soundness:** 3
**Presentation:** 3
**Contribution:** 2
**Rating:** 8
**Confidence:** 3

**Summary:**

This work examines the relationship between the initial Gaussian noise, generated images, and latent representations produced using the DDIM inversion technique in diffusion models.

**Strengths:**

* The paper provides comprehensive empirical evidence to support its claims, using various metrics and visualizations.
* The study demonstrates the limitations of DDIM inversion, particularly its deviation from theoretical expectations and the persistence of inversion errors despite prolonged training.

**Weaknesses:**

* The analysis focuses primarily on DDPM and LDM models, leaving open the question of whether the observed phenomena generalize to other diffusion model architectures.
* While the paper empirically observes the inaccuracy of DDIM inversion and the early formation of noise-to-sample mapping, it lacks a theoretical explanation for these findings.

**Questions:**

* Does the observed early formation of the noise-to-sample mapping have
  connections with the stages of reverse diffusion process as discussed
  in https://arxiv.org/abs/2402.18491?

---

> ### Author Response · Authors · 2024-12-03
> **Response to the Review (1/2)**
>
> We appreciate the feedback and Reviewer's positive opinion about our experiments. We would like to clarify the remaining questions in this comment:
>
> > The analysis focuses primarily on DDPM and LDM models, leaving open the question of whether the observed phenomena generalize to other diffusion model architectures.
>
> To show that our findings generalize over more diffusion models, we performed our experiments also on other large-scale models. To that end, we used an unconditional pixel-space U-Net trained on images from ImageNet with $256\times256$ resolution and a **class-conditional** Diffusion Transformer (DiT) operating in the Latent Space of the autoencoder, trained also on images from ImageNet with $256\times256$ resolution.
>
> First, we include those two models in our experiments to compare pixel correlation in Gaussian noises and latent encodings. The latent encodings created with reverse DDIM for large-scale diffusion models also have correlated pixel values. Surprisingly, the correlation is more significant for the pixel-space model operating at $256\times256$ resolution than for the $64\times64$ model.
>
> |               | DDPM $32\times32$ (CIFAR10) | DDPM $64\times64$ (ImageNet) | DDPM $256\times256$ (ImageNet) | LDM $256\times256$ (CelebA) | DiT $256\times256$ (ImageNet)  |
> |---------------|---------------|---------------|---------------|---------------|---------------|
> | Noise $(x^T)$ | 0.159 ± 0.003 | 0.177 ± 0.007 | 0.141 ± 0.001 | 0.087 ± 0.004 | 0.087 ± 0.004 |
> | Latent $(\hat{x}^T)$ | 0.462 ± 0.009 | 0.219 ± 0.006 | 0.263 ± 0.006 | 0.179 ± 0.008 | 0.171 ± 0.007 |
> | Sample $(x^0)$ | 0.986 ± 0.001 | 0.966 ± 0.001 | 0.985 ± 0.001 | 0.904 ± 0.005 | 0.861 ± 0.004 |
>
>
> Next, we continue this study in the experiment for determining the most probable angles located by the vertexes of images ($x^0$), noises ($x^T$), and latents ($\hat{x}^T$), with varying diffusion steps $T$. We show that, even for large-scale diffusion models, the latents are located along the trajectory of the generated image. Our observations with angles align closely with the correlation experiment.
>
> | Model                             | T    | $\angle x^0$ | $\angle x^T$ | $\angle \hat{x}^T$ |
> |-----------------------------------|------|--------|--------|---------|
> | **U-Net DDPM 32×32**              | 10   | 44     | 16     | 120     |
> |                                   | 100  | 29     | 28     | 123     |
> |                                   | 1000 | 20     | 45     | 115     |
> | **U-Net DDPM 64×64**              | 10   | 30     | 31     | 119     |
> |                                   | 100  | 11     | 60     | 109     |
> |                                   | 1000 | 6      | 79     | 95      |
> | **U-Net DDPM 256×256**            | 10   | 24     | 50     | 106     |
> |                                   | 100  | 24     | 73     | 83      |
> |                                   | 1000 | 23     | 73     | 84      |
> | **U-Net LDM 64×64**               | 10   | 23     | 53     | 104     |
> |                                   | 100  | 2      | 76     | 102     |
> |                                   | 1000 | 1      | 83     | 96      |
> | **DiT LDM 32×32**                 | 10   | 27     | 47     | 106     |
> |                                   | 100  | 4      | 66     | 110     |
> |                                   | 1000 | 1      | 80     | 99      |

---

> > ### Author Response · Authors · 2024-12-03
> > **Response to the Review (2/2)**
> >
> > We also leverage those two models to show that our findings on image-to-noise and noise-to-image mapping by $L_2$-distance are valid for large-scale models. As for previously studied pixel-space diffusion models, we can correctly determine initial noise based on generation $(x^0 \rightarrow x^T)$ by choosing the noise closest to it using the $L_2$-norm. For the $256\times256$ resolution pixel-space model, we obtain $100\%$ accuracy in this assigning. When predicting generation, based on initial noise $(x^T \rightarrow x^0)$, the accuracy is worse than for lower-resolution models. For this particular model, we, once more observed that the reason for such behavior are singular generations with large plain areas that are located close to the mean of the random gaussian noise. For conditional DiT operating in the latent space of the LDM, we show that, similarly to U-Net-based LDM, we can do assignments in both directions (so determining generations based on noises and vice-versa) with an almost $100\%$ success rate, indicating that our findings are valid across variant diffusion architectures and for conditional diffusion models.
> >
> > | T     | DDPM 256×256 $(x^0 \rightarrow x^T)$ | DDPM 256×256 $(x^T \rightarrow x^0)$ | DiT 256×256 $(x^0 \rightarrow x^T)$ | DiT 256×256 $(x^T \rightarrow x^0)$ |
> > |-------|-------------------------------------|-------------------------------------|--------------------------------------------|--------------------------------------------|
> > | 10    | 100 ± 0.0                          | 39.2 ± 6.2                          | 100 ± 0.0                                 | 93.7 ± 7.2                                 |
> > | 50    | 100 ± 0.0                          | 22.9 ± 5.1                          | 100 ± 0.0                                 | 90.8 ± 10.1                                |
> > | 100   | 100 ± 0.0                          | 23.2 ± 4.8                          | 100 ± 0.0                                 | 90.7 ± 10.1                                |
> > | 500   | 100 ± 0.0                          | 25.0 ± 4.6                          | 100 ± 0.0                                 | 93.0 ± 8.5                                 |
> > | 1000  | 100 ± 0.0                          | 25.0 ± 4.4                          | 100 ± 0.0                                 | 96.7 ± 4.6                                 |
> >
> >
> > > Does the observed early formation of the noise-to-sample mapping have connections with the stages of reverse diffusion process as discussed in https://arxiv.org/abs/2402.18491?
> >
> > Thank you for pointing out this insightful related work. In 2402.18491 authors study how noise-to-sample mapping evolves in the backward diffusion process. They show that trajectories can be divided into three regimes where, in the first part, generations of different objects follow common trajectory, then in the second regime they split towards distinct parts of the data distribution, while the third regime corresponds to the memorization and drives samples towards particular data points. In our studies, we focused on how the noise-to-sample mapping change throughout training. We show that general characteristic of the sample is defined early in the training stage what might suggest that the model, early in the training achieves reasonable performance in the second regime defined by Biroli et al., while the third regime is further optimized throughout the remaining training steps. We will add these explanations to the final version of our submission.

---

### Official Review · Reviewer_DKMh · 2024-11-04

**Soundness:** 3
**Presentation:** 2
**Contribution:** 3
**Rating:** 3
**Confidence:** 3

**Summary:**

This paper presents some empirical observations between noise, image, and its inversion: (1) the inversion contains some structure of the original image and is different from the noise; (2) the inversion approximately lies in the trajectory from noise to image; (3) it is possible to assign noise to the corresponding generated images from L2 distance and this mapping is learned at an early stage of training

**Strengths:**

This paper presents intriguing empirical observations, particularly the ability to assign noise to corresponding generated images based on L2 distance. This phenomenon appears to relate to diffusion models and optimal transport, and further exploration could deepen our understanding of diffusion model training dynamics and properties of image manifold.

**Weaknesses:**

The paper feels incomplete to me.

For the empirical observations in Sections 4.2 and 4.3, I don’t see any clear applications based on these findings. I suggest taking it a step further; for example, how could we reduce the divergence between the inversion and the noise? What insight does it provide if the inversion lies along the trajectory from noise to image?

The observation in Section 4.4 is interesting, but the paper doesn’t explore any theoretical insights or potential applications related to this phenomenon. One possible direction could be to link it to optimal transport, building a theoretical framework to better understand the training dynamics of diffusion models.

Overall, without applications or theoretical insights, the empirical analysis lacks clear motivation. I hope the authors can identify relevant scenarios where these observations could be put to practical use.

**Questions:**

1. From Table 2, are there any insights into why LDM could achieve 100% accuracy for both $x^0 \rightarrow x^T$ and $x^T \rightarrow x^0$, but DDPM could only achieve high accuracy for $x^0 \rightarrow x^T$. What key differences between LDM and DDPM might explain this discrepancy?

---

> ### Author Response · Authors · 2024-12-03
> **Response to the Review**
>
> We thank the Reviewer for valuable feedback and valuable suggestions.
>
> > For the empirical observations in Sections 4.2 and 4.3, I don’t see any clear applications based on these findings. I suggest taking it a step further; for example, how could we reduce the divergence between the inversion and the noise? What insight does it provide if the inversion lies along the trajectory from noise to image?
> >
> > Overall, without applications or theoretical insights, the empirical analysis lacks clear motivation. I hope the authors can identify relevant scenarios where these observations could be put to practical use.
>
> Thank you for this suggestion. The main goal of our work was to study a behavior of a commonly used DDIM inversion technique in order to provide explanations and in-dept understanding of its limitations. Agreeing with Prof. Black [https://perceiving-systems.blog/en/post/novelty-in-science], we follow his point of view that a novel paper does not have to come with a direct application. Our insights shed a new light on the problem, show how approximation error influences the results of the reverse DDIM procedure, and how this behavior changes during diffusion training.
>
> > The observation in Section 4.4 is interesting, but the paper doesn’t explore any theoretical insights or potential applications related to this phenomenon. One possible direction could be to link it to optimal transport, building a theoretical framework to better understand the training dynamics of diffusion models.
>
> Thank you for pointing out this interesting direction. There are several works discussing the connection between diffusion models' training dynamics and optimal transport and to the best of our knowledge, this topic is still to be defined. In [1] authors show that DDPM encoder map (e.g. latent-sample mapping) coincides with the optimal transport map when modeling simple distributions. However, as noticed by [2,3], the proof provided by [1] cannot hold. Our experiments show that the closest-L2-based mapping in the case of pixel-space DDPMs holds only in one direction, what might be an interesting starting point for more theoretical considerations. Moreover, we also highlight that this mapping appears relatively early in the diffusion model training, sheding some light on the dynamics of diffusion models' training.
>
> >Questions: From Table 2, are there any insights into why LDM could achieve 100% accuracy for both
> $x^0 \rightarrow x^T$ and $x^T \rightarrow x^0$, but DDPM could only achieve high accuracy for $x^0 \rightarrow x^T$. What key differences between LDM and DDPM might explain this discrepancy?
>
> Thank you for this interesting question. We attribute the difference to the nature of the input data provided to the diffusion in the LDM models. In this scenario the diffusion model is trained on the latent data representations extracted by the autoencoder usually trained with additional regularization either by the KL-Loss or VQ-Loss. In both cases, the application of the regularization leads to the normalization of the input data. As presented in the Appendix, the reason why DDPM does not achieve high accuracy for the $x^T \rightarrow x^0$ case is that there exist some images, and hence, some generations are by nature located closer to the mean of the input data noise, so they ''attract'' more random noises. This is not the case for the LDM model.
>
> **References:**
>
> [1] Khrulkov, Valentin, et al. "Understanding ddpm latent codes through optimal transport." arXiv preprint arXiv:2202.07477 (2022).
>
> [2] Kim, Young-Heon, and Emanuel Milman. "A generalization of Caffarelli’s contraction theorem via (reverse) heat flow." Mathematische Annalen 354.3 (2012): 827-862.
>
> [3] Lavenant, Hugo, and Filippo Santambrogio. "The flow map of the fokker–planck equation does not provide optimal transport." Applied Mathematics Letters 133 (2022): 108225.

---

### Official Review · Reviewer_BpCo · 2024-11-04

**Soundness:** 1
**Presentation:** 1
**Contribution:** 2
**Rating:** 3
**Confidence:** 4

**Summary:**

The authors perform an in-depth analysis of the DDIM inversion technique by analyzing the relationship between initial Gaussian samples, the corresponding generated samples, and their inverted latents. The empirical analysis is presented on pixel and latent space diffusion models for CIFAR-10, ImageNet, and CelebA datasets.

**Strengths:**

The authors attempt to demonstrate the relationship between diffusion latents, their corresponding generated samples, and latents obtained using DDIM inversion. Since DDIM inversion is of interest to practitioners working in controllable synthesis using diffusion models,  some of the analyses presented in the paper in Fig. 4 can be useful.

**Weaknesses:**

Since the paper primarily analyzes the relationship between the diffusion latents, generated samples/ data points, and the reverse DDIM latent without any methodological contributions, I would expect the experiments section to be more detailed and clear. More specifically, the following observations stand out:

1. **Missing experimental details**: For instance, the image resolution at which the models were trained is missing for all datasets. Similarly, details on the network architecture used for the diffusion denoiser and training hyperparameters are missing for both pixel space diffusion models and LDMs. Moreover, it is unclear from the text how the angles between different vectors were computed in Figure 2. While these are only a few examples, I would request the authors include all experimental details in the Appendix.

2. **Limited experiments and overclaiming**: Firstly, if I understand correctly, Sections 4.2 and 4.3 reaffirm already existing conclusions about the DDIM Inversion procedure (as the authors note in lines 172-173) using a different experimental methodology. In this context, can the authors point out additional insights that can be drawn from these experiments? Secondly, the authors note the following in Line 175: We study the implications of this fact and show its far-reaching consequences. However, it is not clear from the main text what these implications are, as these are never discussed and, therefore, seem like overclaiming. Can the authors discuss this in detail? I would have liked to see the impact this can have on the DDIM inversion-based editing or reconstruction capabilities, which would justify this claim. Lastly, I don't see any results on a large-scale experiment (say ImageNet-256), and it is unclear how severe this problem is at scale. Can the authors comment on this and include relevant experiments?

3. While the experiments in Section 4.4 are interesting, what is their significance in the context of the broader picture of the paper? If I understand correctly, the focus of the main text is to highlight issues with DDIM inversion by analyzing the relationship between the inverted latents, the original latents, and the generated samples, and therefore it is not clear how Section 4.4 fits here since there seems to be no reference to DDIM inversion here. Secondly, in Section 4.4, what is the minimum L2 distance criterion for the assignment of images to noise mentioned in line 321?

**Minor Comments**

1. The introduction can be improved. For instance, the authors note the following:
```
Nevertheless, one of the significant drawbacks that distinguishes diffusion-based approaches from other generative models like Variational Autoencoders (Kingma & Welling, 2014), Flows (Kingma & Dhariwal, 2018), or Generative Adversarial Networks (Goodfellow et al., 2014) is the lack of implicit latent space that encodes training data into low-dimensional, interpretable representations.
```
While GANs and VAEs indeed are designed to assign low-dimensional latent codes to the data, Flows/Continuous flows also do not possess a low-dimensional latent space and are similar to diffusion models in that aspect. In fact, the ODE sampling in diffusion models is equivalent to simulating a continuous normalizing flow with a vector field defined in terms of the score function. Therefore, this claim is misleading, and it would be great if the authors could revise this in the main text.

2. **Missing citations**: Reference to related work is missing in some places. For instance, in line 37, `combining diffusion models with additional external models`, references to several related works are missing [1,2]
[1] DiffuseVAE: Efficient, Controllable and High-Fidelity Generation from Low-Dimensional Latents, Pandey et al.
[2] Score-based Generative Modeling in Latent Space, Vahdat et al.

3. Figure 1c: There is a single latent $\hat{x}_T$ for a panel of 4 images, and it is thus confusing. Could the authors clarify which image in this panel the generated latent corresponds to?

4. Table 1: What does each row correspond to? Does it denote the correlation of the pixels in the latent (pure Gaussian noise or reversed DDIM) vector vs data samples?

**Questions:**

See the Weaknesses section.

---

> ### Author Response · Authors · 2024-12-03
> **Response to the Review (1/n)**
>
> We appreciate the Reviewer's valuable feedback.
>
> **Missing experimental details**
> > For instance, the image resolution at which the models were trained is missing for all datasets.
> > Similarly, details on the network architecture used for the diffusion denoiser and training hyperparameters are missing for both pixel space diffusion models and LDMs.
>
> In the experiments for initial submission, we leveraged three diffusion models:
>
> 1. Unconditional pixel-space Denoising Diffusion Probabilistic Model (DDPM), with a U-Net architecture as a backbone. This model was trained on the CIFAR-10 dataset at image resolution $32\times32$. We use the checkpoint from [1]. This model was trained with $T=4000$ diffusion steps, with cosine schedule and hybrid loss (composed of simplified objective and variational lower bound loss). The model was trained for 500K training steps.
> 2. Unconditional pixel-space Denoising Diffusion Probabilistic Model (DDPM), with a U-Net architecture as a backbone, which was trained on the ImageNet dataset with image resolution $64\times64$. We use the checkpoint from [1], which, similarly to (1), was trained with $T=4000$, cosine schedule, and hybrid loss, but for 1.5M training steps.
> 4. Unconditional Latent Diffusion Model (LDM) trained on the CelebA-HQ dataset with images of resolution $256\times256$. This particular model is a U-Net-based denoising diffusion model inside the $3\times64\times64$ latent space of the VQ-VAE autoencoder. We use the trained weights from [2]. The denoising model was trained with $T=1000$ diffusion steps and a linear variance schedule for 410K training steps.
>
> Additionally, as requested, we have added experiments on two additional diffusion architectures focusing on higher resolution data:
>
> 1. Unconditional pixel-space Denoising Diffusion Probabilistic Model (DDPM), with a U-Net architecture as a backbone, that was trained on the ImageNet dataset at image resolution $256\times256$. We use the trained weights from [3]. This model was trained with $T=1000$ diffusion steps and a linear variance schedule for 1980K training steps.
> 2. Conditional Diffusion Transformer (DiT), leveraging Transformer architecture as the denoising diffusion backbone inside the $32\times32\times4$ latent space of Variational Autoencoder, trained on ImageNet dataset with image resolution $256\times256$. For our experiments, we skip the classifier-free guidance. We use the trained weights from [4]. This model was trained with $T = 1000$ diffusion steps and a linear variance schedule for 400K training steps.
>
> For analyzing noise, latent and sample properties with the training progress, we train two unconditional DDPMs with the U-Net architectures - $32\times32$ CIFAR-10 (1) and $64\times64$ ImageNet (2), with the same exact hyperparameters as [1].
>
> > Moreover, it is unclear from the text how the angles between different vectors were computed in Figure 2.
>
> For the experiment with angles (Figure 4), we first sample 1000 example generations, and calculate the angles next to the image $\angle x^0$, the Gaussian noise $\angle x^T$, and the latent encoding $\angle \hat{x}^T$ by calculating the cosine similarity between two vectors attached at a given point and converting this value from radians to degrees.
>
> On top of that, for the visualization in Figure 2, we create histograms for each triangle's vertex and obtain the probability density function for every angle binned up to the precision of one degree. Finally, for all triples of angles that can form a triangle (adding up to 180 degrees), we calculate the probability of such a triangle as the product of the probabilities of each angle. Finally, we visualize the triangles yielding the highest probability.
>
> > Secondly, in Section 4.4, what is the minimum L2 distance criterion for the assignment of images to noise mentioned in line 321?
>
> For calculating a distance between two objects in our experiments, we use the $L_2$ norm of the matrix being a difference between the two objects: ${||x-y||}_2 = \sqrt{\Sigma_c\Sigma_i\Sigma_j(x_{c,i,j}-y_{c,i,j})^2}$.
>
> For the experiment in Table 2, we assume a setup where we have $N$ inputs, called, in the paper, initial Gaussian noises, and their $N$ corresponding diffusion model outputs, called image generations. When assigning images to the noises, we iterate over all the $N$ noises, and, for a particular one, we calculate its distance to all the $N$ generations. We assign the images to noises by choosing those that has the lowest L2 distance.
> For the accuracy metric, we calculate how accurate is the assignment described above. For the reverse problem (assignment of noises to images), the set-up is the same, but we iterate over the $N$ image generations and, for each image, calculate its $L_2$-distance to the noises.

---

> > ### Author Response · Authors · 2024-12-03
> > **Response to the Review (2/n)**
> >
> > > Table 1: What does each row correspond to? Does it denote the correlation of the pixels in the latent (pure Gaussian noise or reversed DDIM) vector vs data samples?
> >
> > In Table 1, we empirically validate that latent representations calculated with reverse DDIM does not follow the random Gaussian distribution. In particular, we calculate correlation between each pair of pixels, and present the average of the correlation coefficients for the top 10 most correlated pairs of pixels. We can observe that latent codes (second row) has at least a small number of highly correlated pixels when compared to the sampled gaussian noise (first row). For completeness, we also present the values for the final generations (third row).
> >
> > > While these are only a few examples, I would request the authors include all experimental details in the Appendix.
> >
> > Thank you for this suggestion, we will gather all of experimental details included in our code repository and add them to the Appendix.
> >
> > > Firstly, if I understand correctly, Sections 4.2 and 4.3 reaffirm already existing conclusions about the DDIM Inversion procedure (as the authors note in lines 172-173) using a different experimental methodology. In this context, can the authors point out additional insights that can be drawn from these experiments?
> >
> > We agree that previous works also indicate that the latent encodings resulting from DDIM inversion are not the same as the initial Gaussian noises. What these works have in common is claiming that latents are not white uncorrelated Gaussians and that they statistically deviate from a normal distribution ([5, 6, 7]). Other works ([8,9]) indicate that the more the latents deviate from Gaussian noise, the worse the quality of images denoised from their interpolation and editing.
> >
> > However, in addition to validating claims made in those papers (see Figure 1 and Table 1), we offer a much more in-depth analysis that highlights the main differences between noises and latents. Specifically, we show that latents are located next to diffusion denoising trajectory, between the initial Gaussian noise and the final images (Figure 2 and Figure 3). Additionally, we show that the inverse DDIM method does not benefit from extending the training of the diffusion model (Figure 4). Finally, through experiments carried out during this Rebuttal, we indicated that, while it is possible to predict, based on the image, which noise it origins from through the $L_2$ distance, we cannot similarly indicate the latent encoding resulting from it through the inverse DDIM process. Therefore, the original noise-to-image mapping does not hold in the latents-to-image scenario.
> >
> > > Secondly, the authors note the following in Line 175: We study the implications of this fact and show its far-reaching consequences. However, it is not clear from the main text what these implications are, as these are never discussed and, therefore, seem like overclaiming. Can the authors discuss this in detail? I would have liked to see the impact this can have on the DDIM inversion-based editing or reconstruction capabilities, which would justify this claim.
> >
> > As noted by the past works ([5,6,8,9]), non-Gaussian properties in the inverted latent encodings lead to the generation of lower quality images and also introduce artefacts into the results of various image manipulations such as interpolation or editing.
> >
> > As our work deeply explores the analysis of the disparity between latents and noise, we believe that the highlighted flaw of preserving image structure in latent encodings is a source of error in all the methods for interpolation and image editing based on the DDIM inversion.
> >
> > > Lastly, I don't see any results on a large-scale experiment (say ImageNet-256), and it is unclear how severe this problem is at scale. Can the authors comment on this and include relevant experiments?
> >
> > We would like to point out that the LDM model we used in the experiments operates on $256\times256$ images from the CelebA dataset. However, the internal diffusion model operates in latent space of $3\times64\times64$. As requested, we performed our experiments also on other large-scale models. To that end, we used (1) an unconditional pixel-space U-Net trained on images from ImageNet with $256\times256$ resolution and a **class-conditional** Diffusion Transformer (DiT) operating in the Latent Space of the autoencoder, trained also on images from ImageNet with $256\times256$ resolution.

---

> > > ### Author Response · Authors · 2024-12-03
> > > **Response to the Review (3/n)**
> > >
> > > First, we include those two models in our experiments to compare pixel correlation in Gaussian noises and latent encodings. The latent encodings created with reverse DDIM for large-scale diffusion models also have correlated pixel values. Surprisingly, the correlation is more significant for the pixel-space model operating at $256\times256$ resolution than for the $64\times64$ model.
> > >
> > > |               | DDPM $32\times32$ (CIFAR10) | DDPM $64\times64$ (ImageNet) | DDPM $256\times256$ (ImageNet) | LDM $256\times256$ (CelebA) | DiT $256\times256$ (ImageNet)  |
> > > |---------------|---------------|---------------|---------------|---------------|---------------|
> > > | Noise $(x^T)$ | 0.159 ± 0.003 | 0.177 ± 0.007 | 0.141 ± 0.001 | 0.087 ± 0.004 | 0.087 ± 0.004 |
> > > | Latent $(\hat{x}^T)$ | 0.462 ± 0.009 | 0.219 ± 0.006 | 0.263 ± 0.006 | 0.179 ± 0.008 | 0.171 ± 0.007 |
> > > | Sample $(x^0)$ | 0.986 ± 0.001 | 0.966 ± 0.001 | 0.985 ± 0.001 | 0.904 ± 0.005 | 0.861 ± 0.004 |
> > >
> > >
> > > Next, we continue this study in the experiment for determining the most probable angles located by the vertexes of images ($x^0$), noises ($x^T$), and latents ($\hat{x}^T$), with varying diffusion steps $T$. We show that, even for large-scale diffusion models, the latents are located along the trajectory of the generated image. Our observations with angles align closely with the correlation experiment.
> > >
> > > | Model                             | T    | $\angle x^0$ | $\angle x^T$ | $\angle \hat{x}^T$ |
> > > |-----------------------------------|------|--------|--------|---------|
> > > | **U-Net DDPM 32×32**              | 10   | 44     | 16     | 120     |
> > > |                                   | 100  | 29     | 28     | 123     |
> > > |                                   | 1000 | 20     | 45     | 115     |
> > > | **U-Net DDPM 64×64**              | 10   | 30     | 31     | 119     |
> > > |                                   | 100  | 11     | 60     | 109     |
> > > |                                   | 1000 | 6      | 79     | 95      |
> > > | **U-Net DDPM 256×256**            | 10   | 24     | 50     | 106     |
> > > |                                   | 100  | 24     | 73     | 83      |
> > > |                                   | 1000 | 23     | 73     | 84      |
> > > | **U-Net LDM 64×64**               | 10   | 23     | 53     | 104     |
> > > |                                   | 100  | 2      | 76     | 102     |
> > > |                                   | 1000 | 1      | 83     | 96      |
> > > | **DiT LDM 32×32**                 | 10   | 27     | 47     | 106     |
> > > |                                   | 100  | 4      | 66     | 110     |
> > > |                                   | 1000 | 1      | 80     | 99      |
> > >
> > > We also leverage those two models to show that our findings on image-to-noise and noise-to-image mapping by $L_2$-distance are valid for large-scale models. As for previously studied pixel-space diffusion models, we can correctly determine initial noise based on generation $(x^0 \rightarrow x^T)$ by choosing the noise closest to it using the $L_2$-norm. For the $256\times256$ resolution pixel-space model, we obtain $100\%$ accuracy in this assigning. When predicting generation, based on initial noise $(x^T \rightarrow x^0)$, the accuracy is worse than for lower-resolution models. For this particular model, we, once more observed that the reason for such behavior are singular generations with large plain areas that are located close to the mean of the random gaussian noise. For conditional DiT operating in the latent space of the LDM, we show that, similarly to U-Net-based LDM, we can do assignments in both directions (so determining generations based on noises and vice-versa) with an almost $100\%$ success rate, indicating that our findings are valid across variant diffusion architectures and for conditional diffusion models.
> > >
> > > | T     | DDPM 256×256 $(x^0 \rightarrow x^T)$ | DDPM 256×256 $(x^T \rightarrow x^0)$ | DiT 256×256 $(x^0 \rightarrow x^T)$ | DiT 256×256 $(x^T \rightarrow x^0)$ |
> > > |-------|-------------------------------------|-------------------------------------|--------------------------------------------|--------------------------------------------|
> > > | 10    | 100 ± 0.0                          | 39.2 ± 6.2                          | 100 ± 0.0                                 | 93.7 ± 7.2                                 |
> > > | 50    | 100 ± 0.0                          | 22.9 ± 5.1                          | 100 ± 0.0                                 | 90.8 ± 10.1                                |
> > > | 100   | 100 ± 0.0                          | 23.2 ± 4.8                          | 100 ± 0.0                                 | 90.7 ± 10.1                                |
> > > | 500   | 100 ± 0.0                          | 25.0 ± 4.6                          | 100 ± 0.0                                 | 93.0 ± 8.5                                 |
> > > | 1000  | 100 ± 0.0                          | 25.0 ± 4.4                          | 100 ± 0.0                                 | 96.7 ± 4.6                                 |

---

> > > > ### Author Response · Authors · 2024-12-03
> > > > **Response to the Review (4/n)**
> > > >
> > > > > While the experiments in Section 4.4 are interesting, what is their significance in the context of the broader picture of the paper? If I understand correctly, the focus of the main text is to highlight issues with DDIM inversion by analyzing the relationship between the inverted latents, the original latents, and the generated samples, and therefore it is not clear how Section 4.4 fits here since there seems to be no reference to DDIM inversion here.
> > > >
> > > > The main aim of our work is to investigate the relationships between the noise space, the image generations produced by the implicit sampler, and the space of the latent encodings resulting from the inversion of the generative process. However, we agree with the Reviewer that the experiment in Section 4.4 could also be used to compare the process of generating images from Gaussian noises with the inversion procedure using the DDIM. To this end, we perform the $L_2$-distance-based assignment experiment, but instead of Gaussian noise, we try to assign image generations $x^0$ to their latent encodings $\hat{x}^T$ and vice-versa. Note that image generations in this scenario are not the reconstructions produced by denoiser from latents but are the same exact generation as in the noise-sample assignment.
> > > >
> > > > First, we assign image generations based on latent encodings $(\hat{x}^T \rightarrow x^0)$. We observe similar results for all five evaluated diffusion models, that align with the previous noise-to-image analysis. Assigning images to their latent encodings in the pixel-space models cannot be successfully done through a simple $L_2$-distance.
> > > >
> > > > | T     | U-Net DDPM 32×32 | U-Net DDPM 64×64 | U-Net DDPM 256×256 | U-Net LDM 256×256 | DiT LDM 256×256 |
> > > > |-------|------------------|------------------|--------------------|-------------------|-----------------|
> > > > | 10    | 38.2 ± 5.1 | 100.0 ± 0.0 | 30.8 ± 4.3 | 100 ± 0.0 | 95.1 ± 6.4  |
> > > > | 100   | 33.4 ± 2.7 | 57.5 ± 7.3  | 23.9 ± 5.0 | 100 ± 0.0 | 90.7 ± 10.3 |
> > > > | 1000  | 40.9 ± 2.7 | 44.7 ± 6.5  | 25.4 ± 4.4 | 100 ± 0.0 | 96.6 ± 4.6  |
> > > > | 4000  | 41.9 ± 3.0 | 43.5 ± 6.5  |      -     |     -     |      -      |
> > > >
> > > > Likewise, we perform an opposite-direction assignment where we try to determine latent encodings based on image generations $(x^0 \rightarrow \hat{x}^T)$. In such a case, surprisingly, for pixel-space models, the results are opposite to the distance-based classification calculated between noises and images, as we cannot assign the correct latent encoding given the distance from the original generation.
> > > >
> > > > | T     | U-Net DDPM 32×32 | U-Net DDPM 64×64 | U-Net DDPM 256×256 | U-Net LDM 256×256 | DiT LDM 256×256 |
> > > > |-------|-----------------------------------------|-----------------------------------------|-----------------------------------------|-----------------------------------------|--------------------------------------------------|
> > > > | 10    | 66.4 ± 1.7 | 64.4 ± 7.1 | 0.7 ± 0.2 | 100 ± 0.0 | 99.8 ± 0.6  |
> > > > | 100   | 16.4 ± 6.1 | 8.6 ± 9.3  | 4.1 ± 1.4 | 100 ± 0.0 | 99.5 ± 1.7 |
> > > > | 1000  | 3.6 ± 2.2  | 1.7 ± 1.3  | 23.9 ± 5.2 | 100 ± 0.0 | 100.0 ± 0.0  |
> > > > | 4000  | 2.8 ± 2.2  | 1.9 ± 1.4  |      -     |     -     |      -      |
> > > >
> > > > We observe for both Gaussian noises $x^T$ and latent encodings $\hat{x}^T$, that the assignment in both directions is possible for Latent Diffusion Models, where the denoising process is performed in the latent space. We hypothesize that this fact is connected with the Kullback-Leibler regularization that imposes a slight KL-penalty towards a standard normal distribution $\mathcal{N}(0, I)$ on the learned latent [2].
> > > >
> > > > > The introduction can be improved. For instance, the authors note [...] While GANs and VAEs indeed are designed to assign low-dimensional latent codes to the data, Flows/Continuous flows also do not possess a low-dimensional latent space and are similar to diffusion models in that aspect. In fact, the ODE sampling in diffusion models is equivalent to simulating a continuous normalizing flow with a vector field defined in terms of the score function. Therefore, this claim is misleading, and it would be great if the authors could revise this in the main text.
> > > >
> > > > Thank you for pointing out this misleading claim. We can see that this statement indeed introduced a lot of confusion, therefore we will remove it from our submission.
> > > >
> > > > > Missing citations: Reference to related work is missing in some places. For instance, in line 37, combining diffusion models with additional external models, references to several related works are missing: (1) DiffuseVAE: Efficient, Controllable and High-Fidelity Generation from Low-Dimensional Latents, Pandey et al. and (2) Score-based Generative Modeling in Latent Space, Vahdat et al.
> > > >
> > > > Thank you for suggesting the missing related works, we will gladly add them to the mentioned section.

---

> > > > > ### Author Response · Authors · 2024-12-03
> > > > > **Response to the Review (5/n)**
> > > > >
> > > > > > Figure 1c: There is a single latent
> > > > >  for a panel of 4 images, and it is thus confusing. Could the authors clarify which image in this panel the generated latent corresponds to?
> > > > >
> > > > > Figure 1c shows the grid of the four latent encodings obtained using the DDIM inversion process from the images shown at left of the figure. We apologise for the confusion, but in the initial submission, we did not add a black border around the latents to easily distinguish them. We are thankful for this remark.
> > > > >
> > > > >
> > > > > **References:**
> > > > >
> > > > > [1] Nichol, Alexander Quinn, and Prafulla Dhariwal. "Improved denoising diffusion probabilistic models." International conference on machine learning. PMLR, 2021.
> > > > >
> > > > > [2] Rombach, Robin, et al. "High-resolution image synthesis with latent diffusion models." Proceedings of the IEEE/CVF conference on computer vision and pattern recognition. 2022.
> > > > >
> > > > > [3] Dhariwal, Prafulla, and Alexander Nichol. "Diffusion models beat gans on image synthesis." Advances in neural information processing systems 34 (2021): 8780-8794.
> > > > >
> > > > > [4] Peebles, William, and Saining Xie. "Scalable diffusion models with transformers." Proceedings of the IEEE/CVF International Conference on Computer Vision. 2023.
> > > > >
> > > > > [5] Parmar, Gaurav, et al. "Zero-shot image-to-image translation." ACM SIGGRAPH 2023 Conference Proceedings. 2023.
> > > > >
> > > > > [6] Garibi, Daniel, et al. "ReNoise: Real Image Inversion Through Iterative Noising." arXiv preprint arXiv:2403.14602 (2024).
> > > > >
> > > > > [7] Hong, Seongmin, et al. "On Exact Inversion of DPM-Solvers." Proceedings of the IEEE/CVF Conference on Computer Vision and Pattern Recognition. 2024.
> > > > >
> > > > > [8] Bodin, Erik, et al. "Linear combinations of Gaussian latents in generative models: interpolation and beyond." arXiv preprint arXiv:2408.08558 (2024).
> > > > >
> > > > > [9] Zheng, PengFei, et al. "NoiseDiffusion: Correcting Noise for Image Interpolation with Diffusion Models beyond Spherical Linear Interpolation." The Twelfth International Conference on Learning Representations. 2024.

---

### Author Response · Authors · 2024-12-03
**General response**

We extremely appreciate the Reviewers' time and the valuable feedback that helped us develop our work. We are thankful for pointing out all the inaccuracies, ambiguities, and errors in the initial submission, which we hope to have addressed in the comments below and which we promise to apply to the final version of the paper. We hope our additional experiments with other diffusion architectures, which we did during the rebuttal, have further strengthened our submission.

---

### Meta-Review · Area_Chair_MpLR · 2024-12-17

**Metareview:**

This paper presents key empirical observations regarding the relationships between noise, images, and their inversions: (1) the inversion retains structural features of the original image and differs from pure noise; (2) the inversion approximately follows the trajectory from noise to the image; (3) it is possible to associate noise with corresponding generated images using L2 distance, and this mapping is learned early in the training process.

However, reviewers find the paper is a bit incomplete, without sufficient experimental evaluation, methods contribution, and explicit practical insights.

**Additional Comments On Reviewer Discussion:**

Although the authors addressed parts of the concerns on experimental evaluation, the practice value of the work remains largely under-explored.

---

### Decision · Program_Chairs · 2025-01-22

Reject